# Extracellular Signal-Regulated Kinases Play Essential but Contrasting Roles in Osteoclast Differentiation

**DOI:** 10.3390/ijms242015342

**Published:** 2023-10-19

**Authors:** Chaekyun Kim

**Affiliations:** BK21 Program in Biomedical Science & Engineering, Laboratory for Leukocyte Signaling Research, Department of Pharmacology, College of Medicine, Inha University, Incheon 22212, Republic of Korea; chaekyun@inha.ac.kr; Tel.: +82-32-860-9874; Fax: +82-32-885-8302

**Keywords:** bone, osteoclast, macrophage colony-stimulating factor (M-CSF), receptor activator of nuclear factor kappa-B ligand (RANKL), mitogen-activated protein kinases (MAPKs), extracellular signal-regulated kinases (ERKs)

## Abstract

Bone homeostasis is regulated by the balanced actions of osteoblasts that form the bone and osteoclasts (OCs) that resorb the bone. Bone-resorbing OCs are differentiated from hematopoietic monocyte/macrophage lineage cells, whereas osteoblasts are derived from mesenchymal progenitors. OC differentiation is induced by two key cytokines, macrophage colony-stimulating factor (M-CSF), a factor essential for the proliferation and survival of the OCs, and receptor activator of nuclear factor kappa-B ligand (RANKL), a factor for responsible for the differentiation of the OCs. Mitogen-activated protein kinases (MAPKs), including extracellular signal-regulated kinases (ERKs), p38, and c-Jun N-terminal kinases, play an essential role in regulating the proliferation, differentiation, and function of OCs. ERKs have been known to play a critical role in the differentiation and activation of OCs. In most cases, ERKs positively regulate OC differentiation and function. However, several reports present conflicting conclusions. Interestingly, the inhibition of OC differentiation by ERK1/2 is observed only in OCs differentiated from RAW 264.7 cells. Therefore, in this review, we summarize the current understanding of the conflicting actions of ERK1/2 in OC differentiation.

## 1. Bone Homeostasis

Bones undergo continuous renewal through the process of bone remodeling throughout an individual’s lifetime. In adult humans, approximately 10% of the total bone content is remodeled every year [1], and this renewal occurs through the destruction and rebuilding of bone at about 1 to 2 million microscopic foci per adult skeleton [2]. The regulation of bone remodeling is highly complex, and bone homeostasis is achieved by the well-balanced actions of bone-forming osteoblasts and bone-destroying osteoclasts (OCs). Osteoblasts are the cells that form new bones, and they also play a central role in the regulation of OC formation and bone resorption through the production of the parathyroid hormone-related protein (PTHrP), macrophage colony-stimulating factor (M-CSF), receptor activator of nuclear factor-κB (NF-κB) ligand (RANKL), and osteoprotegerin (OPG). M-CSF and RANKL are crucial players for OC differentiation and activity. OCs resorb bone by dissolution of the inorganic components and digestion of the bone matrix by secreting hydrogen ions, cysteine proteases, and collagenases. The imbalance between osteoblasts and OCs can lead to skeletal diseases like osteoporosis, osteopetrosis, and rheumatoid arthritis.

## 2. OC Formation and Function

### 2.1. Formation of OCs

OCs originate from common myeloid progenitors that give rise to granulocyte-macrophage colony-forming units (GM-CFU) and then further differentiate into OCs and other cells [3]. The various stages in the formation of OCs include proliferation, maturation, migration, and differentiation of the myeloid progenitors. The differentiation of OCs is induced by two crucial determinants: M-CSF, which is essential for OC survival, and RANKL, which acts as a differentiation factor of OCs [4,5]. The binding of RANKL to its receptor RANK recruits tumor necrosis factor (TNF)-receptor-associated factors (TRAFs), which then activate downstream signaling molecules including mitogen-activated protein kinases (MAPKs), NF-κB, and activator protein-1 (AP-1), leading to the activation of the nuclear factor of activated T cells 1 (NFATc1) [3]. TRAF family proteins such as TRAF2, TRAF3, and TRAF5 bind to RANK and activate the transcription factors including NF-κB, AP-1, and NFATc1 required for OC differentiation, while TRAF6 is the major adaptor protein responsible for the RANKL-induced signaling cascade [6,7]. It is well known that cytokines play a critical role in OC differentiation and the balance between osteoclastogenic and anti-osteoclastogenic cytokines is important for maintaining OC differentiation. Osteoclastogenic cytokines including TNF-α, interleukin-1 (IL-1), -6, -7, -8, -11, -15, -17, -23, and -34, promote OC differentiation, whereas anti-osteoclastogenic cytokines, including interferon (IFN)-α, -β, and -γ, IL-3, -4, -10, -12, -27, and -33 inhibit OC differentiation [8]. The fully differentiated OCs have a short life span, and their survival is promoted by M-CSF, RANKL, IL-1, and calcitonin [6,9,10,11,12]. Estrogen induces apoptosis of OCs and inhibits OC survival [13,14]. Transforming growth factor *β* (TGF-*β*) has both stimulatory and inhibitory effect on OC survival [15,16].

### 2.2. Characteristics of OCs

OCs are characterized as tartrate-resistant acid phosphatase (TRAP)-positive multinucleated cells containing more than three nuclei. RANKL-RANK signaling is crucial in OC formation, as RANK- or RANKL-deficient mice have been found to exhibit abnormal OC development [17,18]. RANKL induces the expression of various genes necessary for OC differentiation and function, such as α_v_β_3_ integrin, TRAP, cathepsin K (CTSK), matrix metalloproteinase 9 (MMP9), OC-associated receptor (OSCAR), calcitonin receptor (CTR), and dendritic cell-specific transmembrane protein (DC-STAMP). TRAP is an iron-containing metalloprotein enzyme expressed in OCs, activated macrophages, and dendritic cells. It plays an essential role in various processes such as skeletal development, collagen processing, bone mineralization, cytokine production by macrophages and dendritic cells, macrophage recruitment, and dendritic cell maturation [19]. TRAP is crucial for maintaining bone and collagen density. TRAP knockout mice exhibit altered collagen synthesis and degradation, reduced bone resorption, and increased bone density [20].

### 2.3. Bone Resorption by OCs

The activation of OCs is characterized by the formation of the ruffled border and the creation of an acidic resorption pit known as Howship’s lacunae. The low pH within the resorption pit dissolves the bone mineral and activates acid proteases, such as cathepsins, that further degrade the organic bone matrix [21]. Initially, OCs adhere tightly to the bone surface and then secrete protons, and proton-activated hydrolytic enzymes into the pit formed between the bone and the OCs [22]. OCs contain many lysosomes which contain TRAP and CTSK, and the formation of the pit occurs through the fusion of lysosomes with the ruffled border [23]. ATPase and carbonic anhydrase II are key players which are responsible for acidification of the resorption fit. Carbonic anhydrase II converts carbon dioxide (CO_2_) to carbonic acid (H_2_CO_3_), which subsequently undergoes ionization to form carbonate and hydrogen ions, and ATPase transfers protons from the cytoplasm into the resorption pit [21,24,25].

## 3. BMM-Derived OCs and RAW 264.7-Derived OCs

Bone marrow cells are efficiently differentiated into OCs in the presence of M-CSF and RANKL. In addition to bone marrow macrophages (BMMs), splenocytes and peripheral blood monocytes can also be differentiated into OCs in the presence of these factors [26,27,28]. Although BMMs are commonly used for OC differentiation in vitro, RAW 264.7 cells, a transformed macrophage-like cell line derived from the lymphoma of a male BALB/c mouse infected with the Abelson murine leukemia virus, have also been known to differentiate into OCs [29,30]. The RAW 264.7 cell is a pure macrophage/OC precursor population that is readily available, easy to culture, and can rapidly differentiate into highly bone-resorptive OC with its hallmark characteristics.

Studies have shown that OCs derived from RAW 264.7 cells (RAW-OCs) exhibit characteristics like those derived from BMMs (BMM-OCs) [31]. RAW 264.7 cell is an immortal and transfectable cell line that exhibits the properties of BMM-OC and is more amenable to genetic manipulation than BMM. Therefore, it is a valuable model for exploring the cellular and molecular regulation of OC differentiation and activation. In a comparative study which examined the steady-state expression levels of mRNA of over 400 genes, the R^2^ value for the pair-wise comparison of BMM-OCs and RAW-OCs was 0.7599, which is similar to those reported in comparisons of the same tissue and cell line analyzed at different times using cDNA array profiling [32]. On the other hand, in a comparative study determined by using quantitative proteomics, the R^2^ value was around 0.13, which indicates low concordance [33].

There are fundamental differences between BMM-OCs and RAW-OCs. However, not many studies comparing these cells have been conducted to date (Table 1). The primary distinction lies in the differentiation requirements of BMMs and RAW 264.7 cells; BMMs need both M-CSF and RANKL to differentiate into OCs, while RAW 264.7 cells can readily differentiate into OCs when treated with RANKL alone [3,31]. According to a proteomics study [33], the expression of OC markers is typically comparable in BMM-OCs and RAW-OCs. However, the weak correlation observed between shared proteins in the two cell types suggests significant differences between them. Additionally, RAW-OCs exhibit prolonged lifespans compared to BMM-OCs. Cheng et al. [34] investigated whether RANKL and M-CSF were essential cytokines for the OC differentiation of RAW 264.7 cells. RANKL increased OC formation and OC marker gene expression, but the addition of M-CSF significantly decreased OC formation and gene expression. Moreover, it was also found that BMM-OCs and RAW-OCs exhibit opposing differentiation responses when subjected to ERK inhibition [35].

## 4. Role of MAPKs in OC Differentiation

MAPKs are evolutionary conserved enzymes connecting cell-surface receptors to critical regulatory targets within cells. They play a vital role in regulating various cellular processes, such as proliferation, differentiation, development, cell cycle, and cell death [43,44]. Mammals express at least four distinct groups of MAPKs, such as extracellular signal-regulated kinase (ERK)1/2, c-Jun N-terminal kinase (JNK)1/2/3, p38α/β/γ/δ, and ERK5. Other MAPKs including ERK3, ERK4, ERK7/8, and Ste20p-related kinases have been discovered [45]. Although MAPKs act in parallel with other cell-signaling systems, intercommunication between the MAPK pathways is very likely, and signal integration transpires at various levels.

The differentiation of BMMs into OCs is regulated by the binding of M-CSF and RANKL to their respective receptors, colony-stimulating factor 1 receptor (CSF1R) and RANK. Both M-CSF and RANKL act through MAPKs during OC differentiation [46,47]. Activation of MAPKs by M-CSF primarily regulates the proliferation of OC precursors [48], and the activation by RANKL is primarily involved in the differentiation of OCs [49], indicating that M-CSF and RANKL have distinct effects on MAPKs. As an example, M-CSF triggers an initial phase of MAPK activation, while RANKL induces both early (5 to 20 min and 1 h) and delayed (8 to 24 h and 24 h) activation of p38 MAPK and ERK in BMMs [39,47]. Therefore, understanding the distinct effects of M-CSF and RANKL on MAPKs leads to a better understanding of OC differentiation.

## 5. Role of ERKs in OC Differentiation

### 5.1. Similarities and Differences between ERK1 and ERK2

ERK signaling is recognized as a critical mechanism by which a variety of extracellular signals regulate cellular functions, including proliferation, differentiation, and cell cycle progression. The ERK isoforms are listed in Table 2. The ERK1 and ERK2 isoforms are highly conserved (84%) kinases that are activated through phosphorylation on threonine and tyrosine residues in the TEY sequence by the dual specificity MAPK kinase MEK [45]. The protein expression level of ERK1 is lower than ERK2 across vertebrates. For example, the total and activated ERK1 were found to be four times less abundant than ERK2 in NIH3T3 cells [50]. The low expression of ERK1 in mice was found to contribute to its faster evolutionary rate [51,52,53].

ERK1 and ERK2 exhibit similar characteristics and functionalities. They share a common docking domain and docking groove, which contribute to similar substrate specificities and activation kinetics [54,55]. They are activated by the same upstream kinase module, and no agonist capable of selectively activating only one of the two kinases has been found [50]. Double silencing of ERK1 and ERK2 is as effective as silencing ERK2 alone in slowing cell proliferation, underscoring the significance of ERK2 in terms of its potency or quantity. However, disruption of ERK1 or ERK2 reveals that the two kinases have isoform-specific functions [45,50,53,56,57,58,59,60]. Mice deficient in ERK1 were found to be viable and exhibited no overt phenotype [61], whereas mice lacking ERK2 were embryonic lethal and ERK1 could not compensate for the loss of ERK2 [62,63,64].

### 5.2. ERK1/2 in OC Differentiation

We assumed that the levels of expression of ERK2 are higher than ERK1 in OCs as in other mammalian tissues [51,52,53]. Although He et al. [65] showed a higher expression level of ERK1 than ERK2 in OC progenitor cells, ERK1 probably has a higher affinity for the antibody compared to ERK2. ERK1/2 are essential for OC differentiation and function. It has been reported earlier that the genetic disruption of ERK1 and ERK2 causes a delay in OC formation by suppressing the RANKL production of osteoblasts [66]. Also, the disruption of ERK1 diminished OC progenitor cell numbers, pit formation, and M-CSF-mediated adhesion and migration, while the disruption of ERK2 reduced OC nucleation and bone resorption without altering OC differentiation, adhesion, and migration [65]. ERK2 could not compensate for the loss of ERK1 in the OCs, whereas ERK1 could compensate for ERK2 disruption. However, these results are contrary to earlier observations where ERK2 shows higher potency and abundance [61,62,63,64]. Taken together, these results suggest that ERK1 and ERK2 play non-redundant isoform-specific functions in OCs (Table 3). 

## 6. The Positive Role of ERK1/2 in OC Differentiation

Activation of ERK1/2 induced by M-CSF mainly promotes the proliferation and survival of OC precursors, and that being triggered by RANKL is responsible for regulating OC formation and function [49]. ERK1 and ERK2 positively regulate the proliferation and differentiation of OCs [39,47,65,66,67]. Moreover, cytokines and growth factors, such as IL-1α, IL-1β, IL-6, IL-15, IL-34, macrophage inflammatory protein-1α (MIP-1α), granulocyte-macrophage colony-stimulating factor (GM-CSF), TNF-α, and fibroblast growth factor-2 (FGF-2) have been known to promote osteoclastogenesis through the activation of ERKs [47,69,70,71,72,73,74,75,76,77,78,79]. Bone morphogenetic protein 9 (BMP9) also positively regulates OC differentiation through ERKs [80]. However, there are conflicting findings regarding the role of cytokines in OC formation. While IL-34 has been shown to activate OC formation [81,82], it is worth nothing that IL-34 has no impact on both in vitro osteoclastogenesis and osteoporosis in ovariectomized rat [83]. The role of IL-35 in osteoclastogenesis is also controversial [84,85]. IL-3, IL-4, TGF-β, osteoactivin, and prostaglandin D2 are known to inhibit OC differentiation by suppressing ERK activation [47,86,87].

Studies suggest that the genetic disruption of ERK1 leads to defects in OC differentiation and function in mice, while disruption of ERK2 does not impair OC differentiation [65]. While ERK1 was largely capable of compensating for ERK2 disruption, ERK2 could not compensate for the loss of ERK1 in OCs. These results indicate that ERK1 plays a more critical role than ERK2 during OC differentiation, despite its relatively low expression [51,52,53]. Pharmacological MEK/ERK inhibitors, such as PD98059, PD0325091, U0126, and FR180204 which inhibited RANKL-induced ERK1/2 phosphorylation, dose-dependently inhibited OC formation [67,86,88]. Moreover, ERK inhibitors significantly inhibited cell migration and in turn cell–cell fusion [39]. PD98059 completely suppressed the fusion and formation of multinuclear cells without affecting the expression of OC markers [39,89]. Moreover, the withdrawal of MEK/ERK inhibitors substantially restored OC formation. The ability of ERKs to positively regulate OC differentiation by suppressing OC apoptosis has also been demonstrated. Dominant negative Ras overexpression has been reported to induce the apoptosis of OCs, and ERK activation by MEK1 prevented apoptosis [90]. PD98059 also promoted the apoptosis of OCs [91].

## 7. The Negative Role of ERK1/2 in OC Differentiation

As described above, it has been shown that ERKs positively regulate OC differentiation and function. However, some findings to the contrary also indicate that ERKs may play a negative role during OC differentiation [35,40,41,42]. Hotokezaka et al. [40] demonstrated that U0126 and PD98059 hindered the differentiation of RAW 264.7 cells into OCs when the cell numbers were low (2000~4000 cells/96-well), but enhanced the differentiation when the cell numbers were high (>8000 cells/96-well). This implies that RAW 264.7 cells have the potential to differentiate into OCs when ERK is inactivated if the cell density is sufficient for contact and fusion to occur. It is not surprising that the inhibition of ERK at low cell numbers reduced OC differentiation due to the lack of cell contact and fusion. However, contrary to the result shown in RAW 264.7 cells, ERK inhibition decreased BMM-OCs despite sufficient cell numbers [35] (our unpublished data). These results suggest that the regulation of OC differentiation by ERK is influenced by factors other than cell numbers. Moreover, U0126 did not alter OC differentiation in RAW 264.7 cells and reversed the decline induced by OPG [41], suggesting a negative role of ERKs in OC differentiation. We also found that ERK inhibitors and ERK1/2-specific small interfering RNA (siRNA) enhanced OC differentiation in RAW 264.7 cells [35,42] (our unpublished data), which confirms that ERK deficiency or inhibition promotes OC differentiation in RAW 264.7 cells.

Although Russo et al. [39] concluded that ERK activates OC differentiation in RAW 264.7 cells, MEK/ERK inhibitors PD98059 and FR180204 increased the expression of OC differentiation markers, such as NFATc1 and TRAP, and TRAF6 and CTSK, respectively. However, it is worth noting that the changes did not reach statistical significance. This suggests that ERK inhibition does not collectively inhibit all molecules involved in OC differentiation but rather stimulates the expression of some of them. This can be explained in part by the findings that showed the different subcellular localizations between phospho-ERKs and OC markers during OC maturation in RAW 264.7 cells [92]. In any case, the enhancement of RANKL-induced OC differentiation through ERK inhibition in RAW 264.7 cells suggests that ERK can play a negative regulatory role in osteoclastogenesis (Figure 1). The negative effect of ERK on hematopoietic cell differentiation is not a new observation. Guihard et al. [58] found that the deletion of ERK1 induced enhanced splenic erythropoiesis, characterized by an accumulation of erythroid progenitors in the spleen. However, the suppression of BMM-OCs by ERK inhibition suggested that ERK also exerts a positive regulatory role during OC differentiation [35].

It is noteworthy that all studies reporting the increase in OC differentiation by ERK inhibitors were conducted in RAW 264.7 cells [35,40,41,42]. The knockdown of the ERK1 or ERK2 gene in RAW 264.7 cells augmented OC differentiation and the expression of OC markers [35]. Although we concluded that ERK exerts a positive regulatory effect on OC differentiation in BMMs and a negative impact on RAW 264.7 cells, numerous studies have shown a positive effect of ERK on OC differentiation in RAW 264.7 cells [72,85,93,94]. The mechanisms by which ERK inhibition leads to an increase in OC differentiation in RAW 264.7 cells remain unclear. It does not seem to be caused by the cell number or the inhibitor itself. MEK/ERK inhibitors PD98059, PD0325091, U0126, and FR180204 possess distinct structures but exhibit a similar ability to enhance OC differentiation in RAW 264.7 cells. As a possible mechanism, we proposed the increase in glutathione (GSH) levels, activation of AMP-activated protein kinase (AMPK), and inhibition of anti-osteoclastogenic factors [35,42]. However, these conditions do not clearly explain how ERKs inhibit OC differentiation. Hence, further investigations are necessary to elucidate the underlying mechanism responsible for the contradictory roles of ERKs in OC differentiation.

## 8. ERK5 in OC Differentiation

ERK5 shares its homology with ERK1/2, but ERK5 does not interact with MEK1 or MEK2 [95]. Nonetheless, the ERK1/2 and ERK5 pathways are responsive to MEK1/2 inhibitors, such as U0126 and PD98059 [96]. Despite their similarities, the targeted deletion of ERK5 in mice has revealed that the ERK5 pathway has distinctive characteristics and plays a critical role in cardiovascular development and endothelial cell function [97,98]. It was found that M-CSF activated ERK5 and OC differentiation [99]. The role of ERK5 in osteoblasts and OCs was not found to be consistent [99,100,101,102]. MEK5/ERK5 inhibitors BIX02189 and XMD 8-92 and MEK5 knockdown inhibited OC differentiation in RAW 264.7 cells [99]. However, the deletion of ERK5 led to severe spinal deformity associated with increased OC activity; the reduction of ERK5 increased OC numbers and expression of OC markers including RANK, CTSK, and NFATc1 in BMMs [103]. Furthermore, the MEK5 inhibitors BIX02188 and BIX02189 stimulated RANKL-induced OC formation in cultures from wild type mice, suggesting that ERK5 negatively regulates OC differentiation.

## 9. p38 MAPK and JNK in OC Differentiation

The p38 MAPK pathway plays a key role in the regulation of OC formation and maturation, and thus, in bone resorption and remodeling [3,68,104,105]. p38 MAPK stimulates the downstream activation of the transcriptional regulator microphthalmia-associated transcription factor (MITF), which controls the expression of the genes encoding TRAP and CTSK [3,106]. The p38 MAPK contains four isoforms of p38 (α, β, γ, and δ) and p38α is the most expressed isoform in OCs that plays a key role in OC differentiation and bone resorption [107]. p38α deficiency has been shown to induce an increase in bone mass, with a decrease in OC numbers and bone resorption [105]. The expression of the dominant negative forms of p38α in BMMs and RAW 264.7 cells results in complete blockage of RANKL-induced osteoclastogenesis [68]. Specific inhibitors of p38 MAPK, PD169316 and SB203580 have been shown to suppress OC differentiation [40,42,104] in RAW 264.7 cells. SB203580 binds to the ATP pocket of the activated p38 MAPK and inhibits phosphorylation of the downstream targets, which leads to the inhibition of OC differentiation. Nevertheless, SB203580 did not inhibit the survival of OC and bone-resorption activity [104]. Although the ERK inhibitor PD98059 did not inhibit OC differentiation in RAW 264.7 cells, SB203580 inhibited the OC differentiation both in BMMs and RAW 264.7 cells [42,68,104]. These results are consistent with our finding that SB203580 inhibits the differentiation in both cells but PD98059 increases the differentiation of RAW 264.7 cells to OCs [35,42].

It has been reported that M-CSF and RANKL activate JNK signaling [108,109]. BMMs isolated from JNK1 knockout mice or carrying a mutated form of c-Jun that cannot be phosphorylated by the JNKs showed a reduction in OC differentiation and bone resorption activity [110]. Blockade of JNK activity at the pre-fusion stage of OC resulted in the reversion of TRAP-positive cells to TRAP-negative cells, even in the continuous presence of RANKL, demonstrating that the JNK pathway is required for maintaining osteoclastic commitment [111]. Moreover, impairment of JNK signaling by the overexpression of dominant-negative JNK1, c-Jun, and c-Fos, or the JNK-specific inhibitor SP600125 abrogated the anti-apoptotic effect of RANKL in OCs [108], indicating that JNK/c-Jun signaling mediates the RANKL-induced anti-apoptotic process in mature OCs. In addition, epidermal growth factor receptor tyrosine kinase inhibitors AG1478 and afatinib suppressed OC differentiation by inhibiting JNK activation, while afatinib slightly increased ERK phosphorylation [112,113]. Taken together, the results indicate that p38 MAPK and JNK positively regulate OC differentiation and function. 

## 10. Conclusions

Remarkable discoveries have been made to broaden the knowledge of the molecular mechanisms involved in the formation and differentiation of OCs. Among the signaling molecules regulating OC differentiation, MAPKs play a critical role. p38 MAPK and JNK positively regulate OC differentiation, whereas ERKs have been reported to regulate OC differentiation positively and negatively. An intriguing point is that the negative regulation of OC differentiation by ERK1/2 either solely through RANKL signaling or in conjunction with both RANKL and M-CSF has been observed exclusively in RAW 264.7 cells. However, there is no evidence that this phenomenon is unique to RAW 264.7 cells.

To better understand this phenomenon, both BMMs and RAW 264.7 cells were treated with various ERK inhibitors, such as PD98059, U0126, and FR180204. The ERK inhibitors suppressed OC differentiation in BMMs but increased it in RAW 264.7 cells in a dose-dependent manner. Moreover, transfection with ERK1 and ERK2 siRNA in RAW 264.7 cells also increased OC differentiation [35,42] (our unpublished data). To our current knowledge, it is difficult to explain the reason for the contrasting results of ERK1/2 in OC differentiation. In fact, there are many papers in which ERK1 and ERK2 have been shown to positively regulate OC differentiation in RAW 264.7 cells. Moreover, many papers have reported that several compounds that affect OC differentiation exert their action through the positive regulatory role of ERK. Hence, elucidating this is a great challenge. The inability to identify a clear mechanism for the role of ERK in OC differentiation can be mainly attributed to the complex nature of the OC differentiation process. Therefore, this review has been unable to provide an answer to the conflicting role of ERKs in OC differentiation. However, this review may raise the right questions and contribute to the research on the role of ERK in the differentiation of OCs.

## Figures and Tables

**Figure 1 ijms-24-15342-f001:**
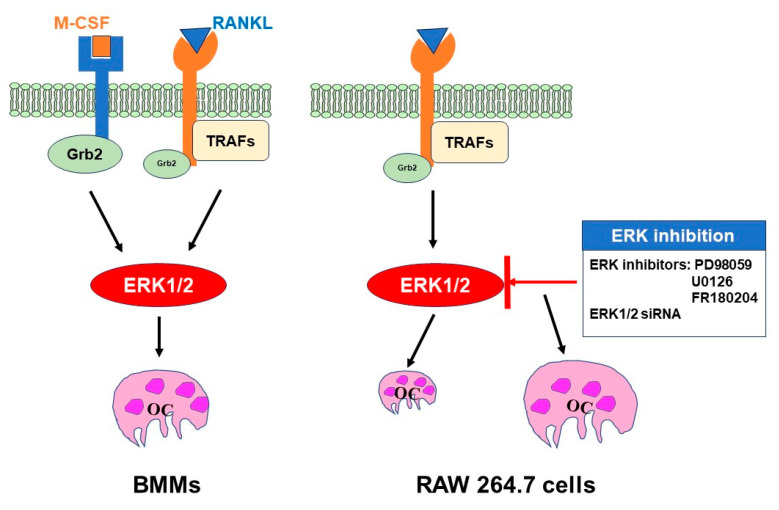
The contrasting roles of ERK1/2 in OC differentiation. ERK1/2 have been reported to regulate OC differentiation positively and negatively. ERK1/2 positively regulate M-CSF and RNAKL-induced OC differentiation in BMMs. However, inhibition of ERK1/2 increases RANKL-induced OC differentiation in RAW 264.7 cells.

**Table 1 ijms-24-15342-t001:** Differences between bone marrow macrophage-derived osteoclast (BMM-OC) and RAW 264.7 cell-derived osteoclast (RAW-OC).

	BMM-OC	RAW-OC	References
v-Abl expression	No	Yes	[29,30]
Cytokine requirement for differentiation	M-CSF and RANKL	RANKL	[3,31,34,35]
Increased gene expression	ACTC1, AP-1, ATP6v0d2, CAH, CARD10, COF2, CTSK, ENDOG, FN1, GIT1, ITGAV, ITGβ3, JUNB, MCL1, MMP9, MYP2, NFATc1, NF-κB2, OSCAR, PXN, RPA2, RRAS, SRC, TPM2, TRAP	ATP6v0d2, AP-1, CAH, CDK6, DAP12, FRA2, GAB2, GSN, ITGAX, JUNB, MCM3 and -5, MMP9, NFATc1, PCNA, POLD1, RELA, SRC, TRAP	[33]
Decreased gene expression	BLNK, CDC2, CDK1, ERK1, FCRγ, c-FMS, IQGAP2, ITGA6, LIG1, MCM2, -3, and -4, NFATc2, NF-κB1, PCNA, POLE3, p90RSK, RELA, RPA1 and -3, TK1, TREM2	BLNK, FCRγ, c-FMS, PI3K	[33]
Apoptosis at 5 days 16 h	78%	42%	[33]
ERK inhibition	inhibits OC differentiation		[35,36,37,38]
		inhibits OC differentiation	[39]
		enhances OC differentiation	[35,40,41,42]

**Table 2 ijms-24-15342-t002:** Isoforms of ERK family members and their inhibitors.

ERK Isoforms		Other Names	Gene Names (Human)	Gene Names (Mouse)	Inhibitors
Typical ERKs	ERK1	MAPK3, p44 MAPK	MAPK3	Mapk3	FR180204, LY3214996 (Temuterkib), PD0325091, PD0325901 (Mirdametinib), PD98059, Selumetinib, U0126
	ERK2	MAPK1, p42 MAPK	MAPK1	Mapk1	FR180204, LY3214996, PD0325091, PD0325901, PD98059, Selumetinib, U0126
	ERK5	BMK1, MAPK7	MAPK7	Mapk7	BIX02188, BIX02188, XMD8-92
Atypical ERKs	ERK3	ERK6, MAPK6, MAPK12	MAPK6, MAPK12	Mapk4	
	ERK4	MAPK4, p63 MAPK	MAPK4	Mapk4	
	ERK7/8	MAPK15	MAPK15	Mapk15 (rat)	

**Table 3 ijms-24-15342-t003:** Effects of ERK1 and ERK2 in RANKL-induced OC differentiation.

ERK Isoforms	Cells	Cytokines	Methods	Effect on OC Formation	OC-Specific Gene Expression	References
ERK1/2	BMMs	M-CSF + RANKL	Gene deletion	ERK1: positive ERK2: ns *		[65]
ERK1/2	BMMs	M-CSF + RANKL	PD98059, U0126	ERK1/2: positive		[35]
ERK1/2	BMMs	M-CSF + RANKL	PD0235901	ERK1/2: positive	ACP5, V-ATPase D2, CTSK, DC-STAMP, c-Fos, NFATc1	[67]
ERK1/2	BMMs	M-CSF + RANKL	PD98059	ERK1/2: no effect		[68]
ERK2	RAW 264.7	RANKL	ERK2 siRNA	ERK2: negative	CTSK, DC-STAMP, NFATc1	[35]
ERK1/2	RAW 264.7	M-CSF + RANKL	PD98059	ERK1/2: no effect		[68]
ERK1/2	RAW 264.7	RANKL	PD98059, U0126	ERK1/2: negative	CTR, CTSK, DC-STAMP, NFATc1, TRAP	[40] [42] [35]
ERK1/2	RAW 264.7	M-CSF + RANKL	U0126	ERK1/2: ns		[41]
ERK1/2	RAW 264.7	RANKL	FR180204	ERK1/2: positive	upregulation: CTSK (ns), TRAF6 (ns) downregulation: DC-STAMP, MMP9, NFATc1, TRAP	[39]
			PD98059		upregulation: NFATc1 (ns), TRAP (ns) downregulation: CTSK (ns), MMP9 (ns)	

* ns: not statistically significant.

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
