# Peer review of "Extracellular Signal-Regulated Kinases Play Essential but Contrasting Roles in Osteoclast Differentiation"

_ijms, 2023, doi:10.3390/ijms242015342_

Round 1

Reviewer 1 Report

The author has offered a comprehensive overview of bone homeostasis, OC formation, and function, as well as the impact of ERK on OC differentiation.

As shown in Table 2, the effects of ERK on OC differentiation display inconsistencies across various papers, encompassing both BMMs and RAW cells. Most of the cited papers are in vitro studies by using inhibitors or siRNA, with one paper [65] conducting in vivo research with genetic mice. It would be valuable for the author to provide their insights on how these in vitro findings relate to in vivo phenotypes.

Additionally, the author has conducted a comparison between BMMs and RAW cells in relation to OC differentiation, as indicated by published studies. Given the varying results obtained from different cell models, it would be advantageous for the author to offer their perspective on selecting the appropriate cell model and experiment system based on the specific objectives of OC research.

The final word "TFG-β" in line 204 should be corrected to "TGF-β" instead of "TFG-β."

Author Response

Dear Reviewer,

We appreciate the careful review and constructive suggestions. Please find enclosed the revised version of the manuscript IJMS-2652108 entitled “Extracellular Signal-Regulated Kinases Play Essential but Contrasting Roles in Osteoclast Differentiation”.

Reviewer 1 Comments:

  1. The final word "TFG-β" in line 204 should be corrected to "TGF-β" instead of "TFG-β."

R: It has been corrected.

Reviewer 2 Report

This is a well written manuscript that primarily details the conflicting roles of ERK1/2 in osteoclastogenesis. The review concisely summarizes the literature with respect to the role MAPKs in osteoclastogenesis with greater depth provided for the conflicting roles of ERK1/2 in RAW vs BMM cell derived osteoclastogenesis.

Suggested changes are listed below.

1.      Include another table listing the ERKs, their other names, their human and mouse genes, and common inhibitors. This would be helpful to the reader.

2.      Line 42: osteoblasts do not secrete PTH but do secrete PTHrP, please correct

3.      Lines 95-96: formatting error, text break

4.      Line 142: change ERK5 to ERKs

5.      Line 147: change c-Fms to more commonly used name CSF1R

6.      Line 165: delete “be” in “to be contribute”

7.      Line 171: the phrase “which suggests the potency or quantity of ERK2” is not a complete thought. Please revise.

8.      Line 181: and should not be italicized.

9.      Line 184: suggest changing to “migration, while the disruption of ERK2….”

10.  Lines 202-203: the phrase “but the additional IL-34…..ovariectomized rat…..” is not clear. Please revise for clarity.

11.  Line 295: please add “(MITF)” after factor and please change CATK to CTSK

suggested edits are provided

Author Response

Dear Reviewer,

We appreciate the careful review and constructive suggestions. Please find enclosed the revised version of the manuscript IJMS-2652108 entitled “Extracellular Signal-Regulated Kinases Play Essential but Contrasting Roles in Osteoclast Differentiation”.

Reviewer 2 Comments:

  1. Include another table listing the ERKs, their other names, their human and mouse genes, and common inhibitors. This would be helpful to the reader.

R: As you suggested, we have included Table 2.

  1. Line 42: osteoblasts do not secrete PTH but do secrete PTHrP, please correct

R: As you suggested, we corrected it.

  1. Lines 95-96: formatting error, text break

R: It has been corrected.

  1. Line 142: change ERK5 to ERKs

R: The suggestion to change ‘ERK5 to ERKs’ has been considered. However, it is important to note that ‘ERK5’ is the established group of the MAPKs, and therefore, we retain ‘ERK5’ in the context. We have modified the following sentence by also including ERK4 and ERK7/8. 

“Mammals express at least four distinct groups of MAPKs, such as extracellular signal-regulated kinase (ERK)1/2, c-Jun N-terminal kinase (JNK)1/2/3, p38α/β/γ/δ, and ERK5. Other MAPKs including ERK3, ERK4, ERK7/8, and Ste20p-related kinases have been discovered [45].”

  1. Line 147: change c-Fms to more commonly used name CSF1R

R: As you suggested, it has been changed.

  1. Line 165: delete “be” in “to be contribute”

R: It has been corrected.

  1. Line 171: the phrase “which suggests the potency or quantity of ERK2” is not a complete thought. Please revise.

R: As you suggested, it has been revised.

  1. Line 181: and should not be italicized.

R: It has been corrected.

  1. Line 184: suggest changing to “migration, while the disruption of ERK2….”

R: As you suggested, it has been changed.

  1. Lines 202-203: the phrase “but the additional IL-34…..ovariectomized rat…..” is not clear. Please revise for clarity.

R: As you suggested, it has been revised.

  1. Line 295: please add “(MITF)” after factor and please change CATK to CTSK

R: As you suggested, it has been changed.